# A Design of a Thermoelectric Energy Harvester for Minimizing Sensor Module Cost

**Kazuma Koketsu and Toru Tanzawa \***

Graduate School of Integrated Science and Technology, Shizuoka University,
Hamamatsu 432-8561, Japan

\*   Correspondence: toru.tanzawa@shizuoka.ac.jp

**Abstract:** This paper discusses a relationship between thermoelectric generator (TEG) electrical parameters, power efficiency of converters, and power consumption of loads in autonomous sensor modules. Based on the method discussed, one can determine the total number of TEG units together with the number of TEG arrays and the number of TEG units connected in series per array when the characteristics of TEG unit, the minimum temperature difference in operation, the power conversion efficiency of the converter and the load condition are given. A practical design flow to minimize TEG cost is proposed and demonstrated, taking the maximum open circuit voltage of TEG and the dependence of the power conversion efficiency of the converter on the input voltage of the converter into consideration. The entire system including TEG and a Dickson charge pump converter, which were designed through the proposed flow, was validated with SPICE.

**Keywords:** thermoelectric generator; converter; load; equivalent circuit model; maximum power point; sensor

## 1. Introduction

A thermoelectric generator (TEG) is a device generating electric power based on temperature differences, which is known as the Seebeck effect [1–3]. A TEG is a key device for energy harvesting among many alternatives such as photovoltaic generators and electrostatic, electromagnetic, magnetostrictive or piezoelectric vibration devices [4]. Given that a nominal TEG can only generate an output voltage on an order of 10–100 mV with a few K temperature difference, a power converter is needed to operate integrated circuits (ICs) including sensor and RF at a higher voltages such as 3 V in autonomous sensor modules [5–9], as shown in Figure 1, where $V_{OC}$ ($R_{TEG}$) is the open circuit voltage (output resistance) of TEG, $\eta$ is the power conversion efficiency of the converter, and $V_{PP}$ ($I_{PP}$, $P_{OUT}$) is the output voltage (average output current, average output power) of the converter to drive sensor and RF blocks.

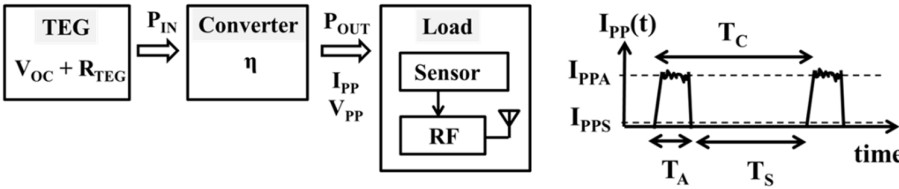

**Figure 1.** Block diagram of an energy harvesting system with TEG, converter, sensor and RF.

Characteristics of the output current ($I_{OP}$) and voltage ($V_{OP}$) of TEGs are described in Figure 2a with an equivalent circuit with $V_{OC}$ and $R_{TEG}$ as shown in Figure 2b, where $I_{SC}$ is

the short circuit current of the TEG and $P_{IN}$ is the output power of the TEG or the input power of the converter.

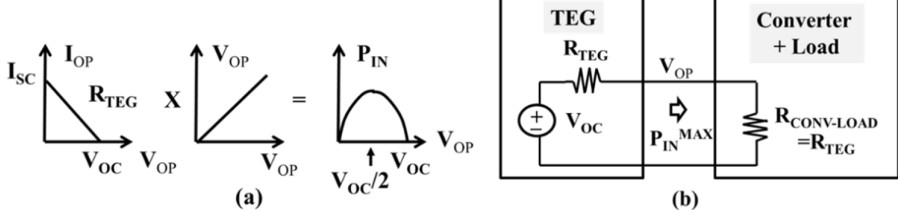

**Figure 2.** (**a**) $I_{OP}$ and $P_{IN}$ of TEG as a function of $V_{OP}$ and (**b**) equivalent circuit under a maximum-output-power condition with impedance matching.

As a result, $P_{IN}$ is described by a parabola where the peak power is given at the interface voltage $V_{OP} = V_{OC}/2$. Based on the equivalent circuit of a TEG system as shown in Figure 2b, the converter is designed to operate TEG at the maximum power point with a given $I_{OP}$ -$V_{OP}$ characteristic of TEG [10,11]. When TEG cannot operate at the maximum power point due to low input voltage, the converter needs to control the input voltage as well as the output voltage [12]. Once the application is determined, the required output current of the converter ($I_{PP}$) can be estimated by using (1),

$$I_{PP} = (I_{PPA}\ T_A + I_{PPS}\ T_S)/T_C \tag{1}$$

where $I_{PPA}$, $T_A$, $I_{PPS}$, $T_S$, and $T_C$ are an average current in operation, an operation period per sense and data transmission, an average stand-by current, a stand-by period, and a cycle time per operation, respectively, as shown in Figure 1. Note that a rechargeable battery or a large capacitor is usually connected at the input terminal of the loading device to stabilize the input voltage of the sensor/RF IC against large $I_{PPA}$. Figure 3 shows the average power as a function of $T_C$ in case of $V_{PP}$ of 3 V, $I_{PPS}$ of 1μA, $I_{PPA}$ of 10 mA, a bit rate of 1 Mbps, and 1 k-bytes/packet with Bluetooth low energy [13]. At a duty of $10^{-4}$ or lower, $I_{PP}$ can be as low as 10 μW. Thus, the requirement for the output power of the converter is determined. From a system viewpoint, one may want to design a TEG structure in such a way that the output power of the converter is maximized under a given load condition.

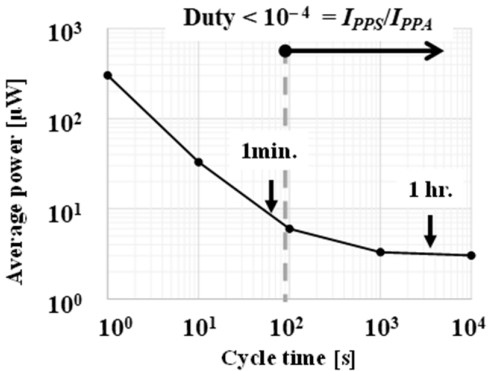

**Figure 3.** Average power of a sensor module as a function of the cycle time.

Table 1 illustrates a TEG composed of multiple pairs of n- and p-type thermocouples (TC). $N_S$ ($N_P$) is the number of TCs connected in series (parallel). In this example, 8 TCs are connected in series (a) or arranged with two arrays of 4 TCs serially connected (b). The former configuration has higher $V_{OC}$ and larger $R_{TEG}$ than the latter does, as shown by (a) and (b) of Figure 4.

**Table 1.** Electrical parameters depending on TEG array structure.

| | $N_S \times N_P$ | TEG array structure | $V_{OC}$ | $R_{TEG}$ | $I_{SC}$ |
|---|---|---|---|---|---|
| (a) | 8 × 1 | T1, T2 | 8 | 8 | 1 |
| (b) | 4 × 2 | T1, T2 | 4 | 2 | 2 |

Thus, even though the area is given, one has a degree of freedom in a combination of $N_S$ and $N_P$ while the multiple of them is constant. In [14], a design technique was proposed to extract the maximum power over a wide $V_{OC}$ range in case of a lack of converter by varying a combination of $N_S$ and $N_P$. However, to the author's knowledge, there have been no design considerations for TEG with converters under given load conditions in the literature to answer the question of how one can determine $N_S$ and $N_P$ under given system conditions. For example, as shown in Figure 4, the operating point given by the cross point of the $V_{OP} - I_{OP}$ curves for the TEG and the converter depends on the slope of the $V_{OP} - I_{OP}$ curve for the converter of smaller (c) or larger (d) than −1. Since TEG is one of the most significant devices in terms of sensor module cost, its size or area must be minimized to enable massively distributed sensor modules.

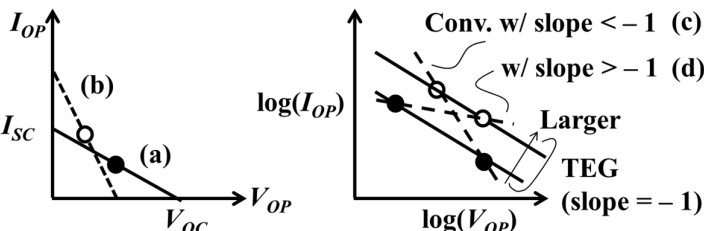

**Figure 4.** $V_{OP} - I_{OP}$ curves of the TEGs (a) and (b) shown in Table 1 and those of the converters whose slope are smaller (c) or larger (d) than −1.

This paper discusses a relationship between TEG electrical parameters, power efficiency of the converter, and power of the load toward minimizing TEG cost. How $V_{OC}$ or $R_{TEG}$ should be determined is shown. In addition, a design flow is proposed to minimize TEG area when the load condition is given, a Dickson charge pump (CP) [15] as converter is used to be integrated in the sensor, with an RF chip as a cost-effective solution.

## 2. Equations between TEG, Converter, and Load

By definition, as described in Figure 1,

$$P_{OUT} = \eta \ P_{IN} \tag{2}$$

To extract power from TEG as much as possible, the converter needs to be operated to match the input impedance of the converter with the output impedance of TEG for impedance matching, as illustrated in Figure 2b. Under the maximum-output-power condition, $P_{IN}$ is given by (3).

$$P_{IN} = (V_{OC}/2)^2/R_{TEG} \tag{3}$$

From (2) and (3), TEG device parameters and circuit parameters are related by (4).

$$V_{OC}^2/R_{TEG} = 4\ V_{PP}\ I_{PP}/\eta \tag{4}$$

$V_{OC}$ is proportional to $\triangle T$ [2]. $V_{OC}$ and $R_{TEG}$ can be varied proportionally by changing TEG structure as described in Table 1. As a result, when specific TCs are characterized, $V_{OC}$ and $R_{TEG}$ are related as in (5).

$$V_{OC} = N_S\ V_{TC},\ R_{TEG} = N_S/N_P\ R_{TC} \tag{5}$$

where $V_{TC}$ and $R_{TC}$ are an open circuit voltage and an output impedance of a TC, respectively. The area of TEG can be estimated by the area of TC ($A_{TC}$) from (5),

$$A_{TEG} = N_S\ N_P\ A_{TC} = (V_{OC}^2/R_{TEG})/(V_{TC}^2/R_{TC})\ A_{TC} \tag{6}$$

$V_{OC}$ can be also shown with $R_{TEG}$, instead of $N_S$ from (5), as below.

$$V_{OC} = (V_{TC}/R_{TC})\ N_P\ R_{TEG} \tag{7}$$

Finally, TEG with minimum area and the maximum operating point are determined by the filled circle rather than the blank one on the curve (c) or (d) in Figure 4, depending on the converter characteristic with a slope of <−1 or >−1. A trajectory of the maximum power point of TEG with a given area on $\log(I_{OP}) − \log(V_{OP})$ plane has a slope of −1. The trajectory of smaller TEG becomes closer to the origin. When the converter has a slope of <−1 as described by the curve (c) in Figure 4, the maximum power point is located at a relatively higher $V_{OP}$ and a relatively lower $I_{OP}$ than the case of using a converter whose slope is greater than −1.

Several conditions for TEG design are studied as follows. When the TEG area and structure are given, $V_{OC}$ can be varied only by increasing $\triangle T$. The minimum $\triangle T$ is determined by (4). $V_{OC}$ depends on the square root of $V_{PP}$, $I_{PP}$, $R_{TEG}$, and $\eta$. Among them, $V_{PP}$ and $\eta$ are expected to not change significantly, at least in a short term. Figure 5 shows $V_{OC}$ vs. $\eta$ with $I_{PP} = 30\mu A$ or 3 $\mu A$ and $R_{TEG} = 300\Omega$ or 1 k$\Omega$ at $V_{PP} = 3$ V based on (4). When $\eta$ is nominally 50%, an improvement in $\eta$ by 10% only gives 10% reduction in $V_{OC}$. Similar goes to $V_{PP}$. As a result, it is considered that $V_{PP}$ and $\eta$ are not effective design parameters to mitigate the requirement for reducing $V_{OC}$.

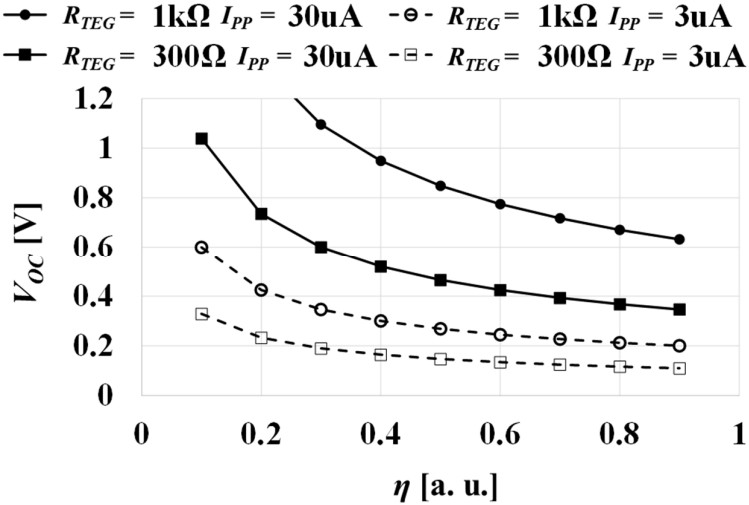

**Figure 5.** $V_{OC}$ vs. $\eta$ with $I_{PP} = 30$ µA or 3 µA and $R_{TEG} = 300$ Ω or 1 kΩ at $V_{PP} = 3$ V.

On the other hand, when applications allow 10X longer cycle time as shown in Figure 3, the required $V_{OC}$ can be significantly reduced, resulting in reduction in TEG cost with reduced $N_S$. Next, let's look at the relationship between $V_{OC}$ and $R_{TEG}$ when $\eta$ and the load condition are assumed. Figure 6 shows $V_{OC}$ vs. $R_{TEG}$ with different $I_{PP}$, $\eta = 0.5$, $V_{PP} = 3$ V,

based on (4). If $R_{TEG}$ needs to increase for small form factor by a factor of 10, $V_{OC}$ has to increase by a factor of 3.2. Alternately, if $T_C$ can be relaxed by a factor of 10 by reducing the frequency of sense and data transmission to 1/10 in a certain application, $I_{PP}$ can decrease by a factor of 10, which allows the system to work with $V_{OC}$ unchanged.

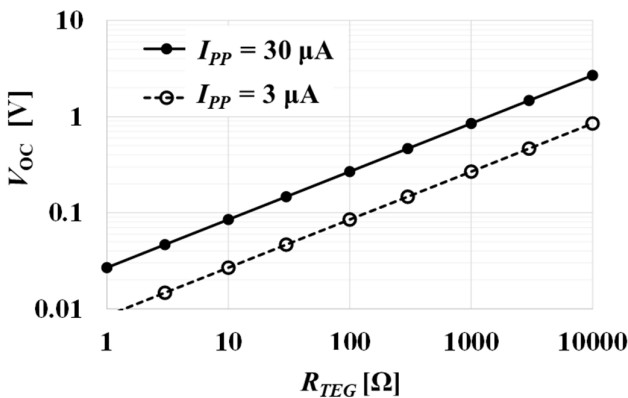

**Figure 6.** *$V_{OC}$* vs. *$R_{TEG}$* with different *$I_{PP}$*, $\eta = 0.5$, *$V_{PP}$* = 3 V.

How can one determine $R_{TEG}$ when $V_{OC}$ is limited by the minimum operation voltage of the converter $V_{DD}{}^{MIN}$? Figure 7 shows $R_{TEG}$ vs. $I_{PP}$ with $V_{OC} = 0.4$ V or 0.8 V, $\eta = 0.5$, $V_{PP} = 3$ V. Even if $V_{DD}{}^{MIN}$ of the converter can be reduced from $V_{OP} = V_{OC}/2 = 0.4$ V in case of $V_{OC} = 0.8$ V to $V_{OP} = 0.2$ V with converter designers' effort, $R_{TEG}$ also has to be reduced by a factor of 4 with the same $\triangle T$ and $I_{PP}$, or $I_{PP}$ also has to be reduced by a factor of 4 with the same $\triangle T$ and $R_{TEG}$, instead. Thus, the effort of improving the converter with respect to reduction in $V_{DD}{}^{MIN}$ requires more effort of reducing $R_{TEG}$ for TEG designers or of reducing $I_{PP}$ for system designers.

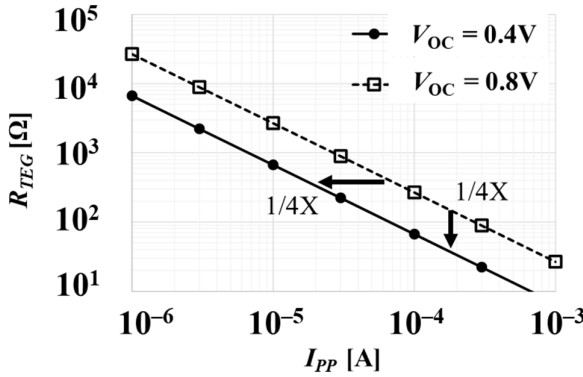

**Figure 7.** *$R_{TEG}$* vs. *$I_{PP}$* with $V_{OC} = 0.4$ V, 0.8 V, $\eta = 0.5$, *$V_{PP}$* = 3 V.

Figure 8 shows a relationship between (4) and (7). The cross points of them express the values of $R_{TEG}$ and $V_{OC}$ for a given condition of $V_{TC}/R_{TC} = 0.45$ mA, $I_{PP} = 30$ μA, $\eta = 0.5$, $V_{PP} = 3$ V. Given that $\eta$ is assumed to be constant over $V_{OP}$ for simplicity in this section, one cannot determine $N_S$ and $N_P$ to minimize TEG area. Therefore, in order to design TEG with minimum cost, a converter needs to be optimally designed by $V_{OP}$ precisely.

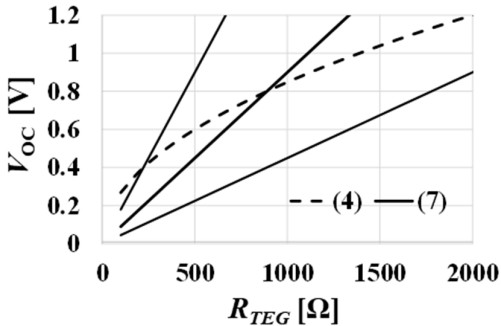

**Figure 8.** $V_{OC}$-$R_{TEG}$ curves for TEG/converter power condition (4) and TEG characteristics (7) when $V_{TC}$/$R_{TC}$ = 0.45 mA, $I_{PP}$ =30 µA, $\eta$ = 0.5, $V_{PP}$ = 3 V.

## 3. Design Flow of TEG with Minimum Area

In the above section II, $\eta$ was assumed to be constant to overview the relationship between TEG electrical characteristics, converter power efficiency, and the load condition of the sensor module. In this section, a more practical design flow is proposed to determine both $N_S$ and $N_P$ of TEG and the design parameters of CP whose $\eta$ can vary as $V_{OP}$ at the same time.

(Assumption) The following parameters are given: $V_{TC}$, $R_{TC}$, and the target $I_{PP\_TGT}$ at $V_{PP}$.

(Parameters to be determined) $N_S$, $N_P$, in such a way that TEG area, i.e., the product $N_S N_P$, is minimum, as well as the number of stage $N_{CP}$, capacitance per stage $C_{CP}$ and clock frequency $f_{CP}$ to design CP.

(Step 1) Design CP with the maximum power conversion efficiency for each $V_{OP}$ when the target $I_{PP}$ is given at a specific $V_{PP}$, based on [16] as below.

It is assumed that (1) CP to be designed is a Dickson type [15], (2) it operates in slow switching limit (SSL) where the clock frequency is low enough to transfer the charges from one stage to the next one through a switching MOSFET in the subthreshold region or namely a switching diode, a unit of the diode has a voltage($V_D$)–current($I_D$) relationship specified by (8), and the oscillator cell consumes much lower power than the CP. Design flow in fast switching limit is open for the future work.

$$I_D = I_S e^{V_D/V_T} \tag{8}$$

The output voltage($V_{OUT}$)–current($I_{OUT}$) relationship of the CP is given by (9) where the output impedance $R_{PMP}$ and the maximum attainable voltage $V_{MAX}$ are given by (10) and (11), respectively. The top plate parasitic capacitance $\alpha_T$ is assumed to be given by (12), where $N_D$, $A_D$, and $C_J$ are the number of unit diodes, the junction area of a unit diode, and the junction capacitance of a unit diode. $V_{TH}^{EFF}$ is an effective threshold voltage given by (13) [17], which is defined by the voltage difference between the adjacent capacitors at the negative clock edge, indicating the voltage loss per stage.

$$I_{OUT} = (V_{MAX} - V_{OUT})/R_{PMP} \tag{9}$$

$$R_{PMP} = \frac{N_{CP}}{f_{CP}C_{CP}(1 + \alpha_T)} \tag{10}$$

$$V_{MAX} = \left(\frac{N_{CP}}{1+\alpha_T} + 1\right) V_{OP} - (N_{CP} + 1)\ V_{TH}^{EFF} \tag{11}$$

$$\alpha_T = N_D A_D C_J / C_{CP} \tag{12}$$

$$V_{TH}^{EFF} = V_T \ln(4^{\frac{1}{N_{CP}+1}} \frac{(1 + \alpha_T)f_{CP}C_{CP}V_T}{N_D A_D I_S}) + \frac{N_{DA_D}I_S}{2f_{CP}C_{CP}(1 + \alpha_T)} \tag{13}$$

The input current $I_{OP}$ of the CP is given by (14) as a function of the output current $I_{PP}$ and the input voltage $V_{OP}$. The last term comes from the reverse leakage of switching diodes.

$$I_{OP} = \left(\frac{N_{CP}}{1 + \alpha_T} + 1\right)I_{PP} + \left(\frac{\alpha_T}{1 + \alpha_T} + \alpha_B\right)N_{CP}f_{CP}C_{CP}V_{OP} + \frac{N_{CP}N_{DA_D}I_S}{2} \tag{14}$$

The power conversion efficiency is defined by (15).

$$\eta = \frac{V_{PP}I_{PP}}{V_{OP}I_{OP}} \tag{15}$$

The optimum number of stages $N_{OPT}$ to maximize the power efficiency is estimated by (16) using the minimum number of stages to output $V_{PP}$ with zero output current given by (17) [18,19], where [X] indicates a rounded integer number of X.

$$N_{OPT} = [1.4 N_{MIN}] \tag{16}$$

$$N_{MIN} = \frac{V_{PP} - V_{OP} + V_{TH}{}^{EFF}}{V_{OP}/(1 + \alpha_T) - V_{TH}{}^{EFF}} \tag{17}$$

CP design flow starts with an initial condition on the target $I_{PP\_TGT}$ at $V_{PP}$, $V_{OP}$, CP area $A_{CP}{}^{INIT}$. $I_{PP}$ and $V_{PP}$ are specified by the loading devices such as sensor and RF ICs. The goal is determining the TEG configuration and the circuit parameters of the CP such that TEG and CP areas are minimized.

Consequently, $N_D$ and $V_{TH}{}^{EFF}$ are treated as variables. One can calculate the flowing parameters step by step: $N_{MIN}$ by (17), $N_{OPT}$ by (16), $C_{CP}$ by (18), and $\alpha_T$ by (12). It is assumed in (18) that the CP area is occupied by the capacitors and switching diodes, where $C_{OX}$ is the capacitance density of each capacitor.

$$C_{CP} = (A_{CP}{}^{INIT}/N_{OPT} - (1 + 1/N_{OPT})N_D A_D)C_{OX} \tag{18}$$

One can numerically solve (13) for $f_{CP}$ because the remaining parameters are determined. From (10) and (11), $R_{PMP}$ and $V_{MAX}$ are calculated. Then, $I_{PP}$ is determined by (19).

$$I_{PP} = (V_{MAX} - V_{PP})/R_{PMP} \tag{19}$$

When $I_{PP}$ is not equal to $I_{PP\_TGT}$, $C_{CP}$ and $N_D$ need to be scaled up or down by the scaling factor $S_F$ given by (20). When both $C_{CP}$ and $N_D$ are scaled proportionally, the optimum $f_{CP}$ can stay the same value because (13) has $C_{CP}$ and $N_D$ only as their ratio. Thus, the required CP area to output $I_{PP\_TGT}$ at $V_{PP}$ is determined by (21).

$$S_F = I_{PP\_TGT}/I_{PP} \tag{20}$$

$$A_{CP} = S_F A_{CP}{}^{INIT} \tag{21}$$

This flow can be done with various combinations of $V_{TH}{}^{EFF}$ and $N_D$. One can determine the best combination of all the CP parameters such as $V_{TH}{}^{EFF}$, $N_D$, $N_{CP}$, $C_{CP}$, and $f_{CP}$ to have the maximum $\eta$ for a given $V_{OP}$. One then needs to repeat the above procedure for various $V_{OP}$. The resultant $V_{OP} - I_{OP}$ and $V_{OP} - A_{CP}$ curves will be used together with those for TEG to determine the target configurations of TEG and CP with minimum areas as presented below.

(Step 2)

2-1: When $V_{OP} < V_{OC\_MAX}/2$ where $V_{OC\_MAX}$ is $V_{OC}$ with $N_P = 1$, find the operating point $(V_{OP}, I_{OP})$ in such a way that $V_{OC} = 2\ V_{OP}$ and $R_{TEG} = V_{OP}/(2\ I_{OP})$ which meets the maximum power condition (3), as shown by the line (a) in Figure 9.

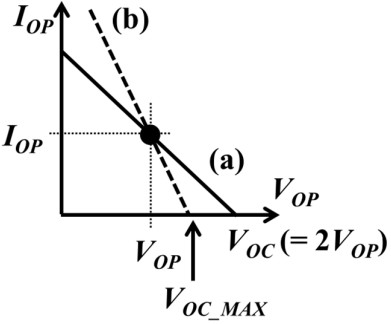

**Figure 9.** Operating point of TEG/Conv. vs. $V_{OC}$ depending on $V_{OC\_MAX}$.

Hence, one can determine

$$N_S = 2\ V_{OP}/V_{TC},\ N_P = 4\ R_{TC}I_{OP}/V_{TC} \tag{22}$$

Then, $A_{TEG}$ is estimated by (23), based on (6).

$$A_{TEG} = (8\ V_{OP}I_{OP})/(V_{TC}^2/R_{TC})\ A_{TC} \tag{23}$$

2-2: When $V_{OP} > V_{OC\_MAX}/2$, one cannot design TEG to run at the maximum operating point even with $N_P = 1$, as shown by the line (b) in Figure 9. Instead, TEG needs to have the following parameters:

$$V_{OC} = V_{OC\_MAX} = N_S\ V_{TC},\ R_{TEG} = (V_{OC\_MAX} - V_{OP})/I_{OP} \tag{24}$$

Then, $A_{TEG}$ is estimated by (25), based on (6).

$$A_{TEG} = (V_{OC\_MAX}/V_{TC})\ A_{TC} \tag{25}$$

where $N_S$ and $N_P$ are given by (26).

$$N_S = 2\ V_{OP}/V_{TC},\ N_P = 1 \tag{26}$$

(Step 3) Find $V_{OP}$ to minimize $A_{TEG}$ among the values found in Step 2 in the $V_{OP}$ range. One can also determine the design parameters of CP such as $N_{CP}$, $C_{CP}$, and $f_{CP}$ at the same time.

Let's see how the above flow works using the parameters in Table 2, which were presented in [16], for demonstration.

**Table 2.** Design and device parameters for demonstration.

| Parameter | Symbol | Value |
|---|---|---|
| Output voltage of CP | $V_{PP}$ [V] | 3.0 |
| Output target current of CP | $I_{PP\_TGT}$ [μA] | 30 |
| Thermal voltage of switching diodes | $V_T$ [mV] | 25 |
| Saturation current density of the diodes | $I_S$ [nA/μm²] | 0.1 |
| Junction capacitance density of the diodes | $C_J$ [fF/μm²] | 3.5 |
| Capacitance density of CP capacitors | $C_{OX}$ [fF/μm²] | 10 |
| Junction area of a unit diode | $A_D$ [μm²] | 10 |
| Bottom plate parasitic cap ratio to the CP cap | $\alpha_B$ [a.u.] | 0.1 |

Figure 10a–e show $\eta$ vs. CP area when $V_{TH}^{EFF}$ is varied between 0.02 V and 0.15 V and $N_D$ is varied among 10, 30, 100, 300, 1000 at $V_{OP}$ of 1.25 V in (a) through 0.25 V in (e), respectively. In this work, $\eta$ is the highest priority, but a very strict constraint could need too large a CP area. Considering a trade-off between $\eta$ and CP area, the best combination of the CP design parameters is determined, in order to have 2% lower $\eta$ than its peak value, which is shown by an arrow in each figure. There were two groups in Figure 10b.

One has $\eta > 0.55$ and the other has $\eta < 0.5$. The former has $N_{CP}$ of three whereas the latter has $N_{CP}$ of four. As $V_{OP}$ decreases, the number of groups with different numbers of $N_{CP}$ increases. Smooth variations on $\eta - CP$ area curves come from variations in $V_{TH}^{EFF}$ or $N_D$ while $N_{CP}$ is unchanged.

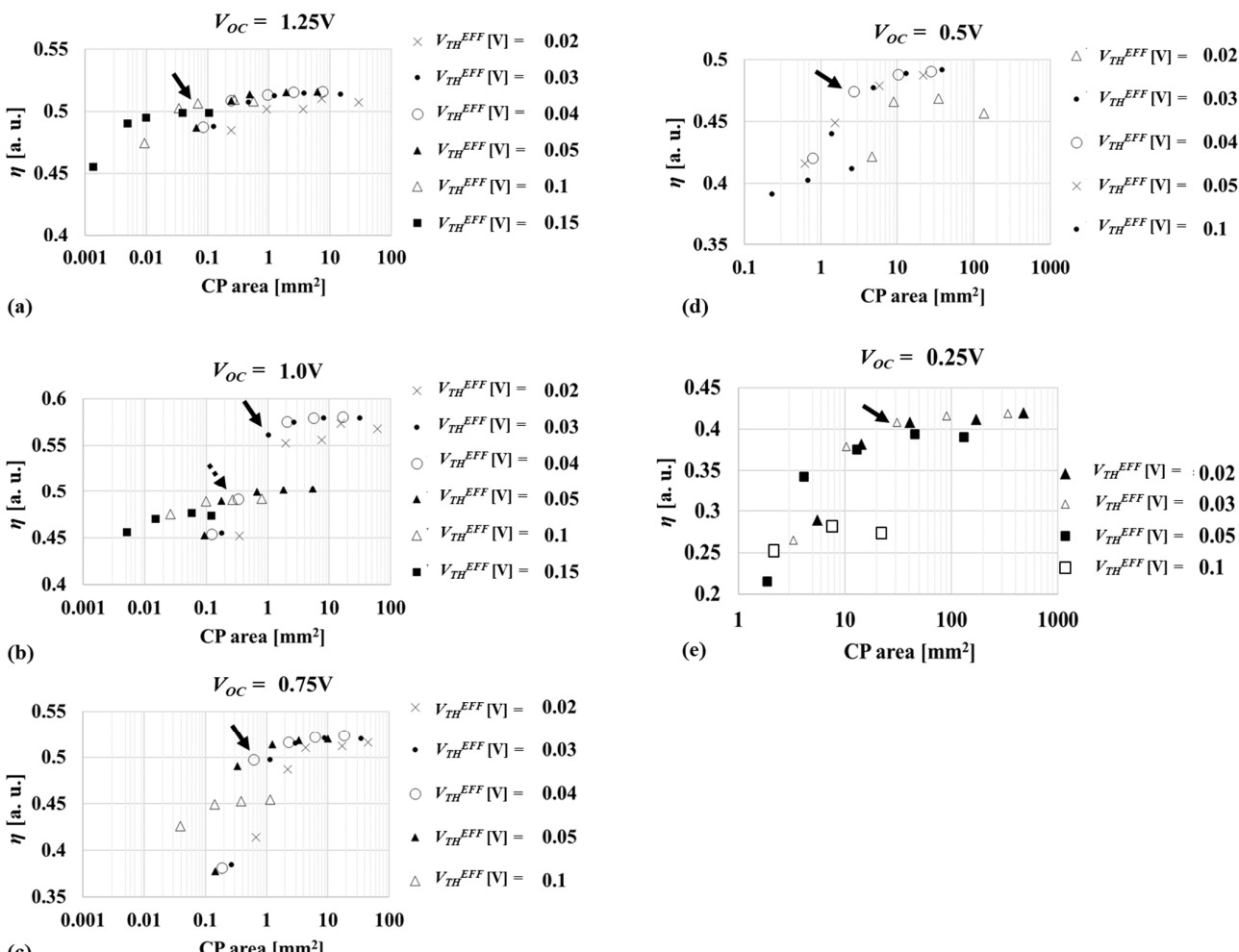

**Figure 10.** $\eta$ vs. CP area at $V_{OP}$ at 1.25 V (**a**), 1.0 V (**b**), 0.75 V (**c**), 0.5 V (**d**), and 0.25 V (**e**). $V_{TH}^{EFF}$ is varied between 0.02 V and 0.15 V and $N_D$ is varied among 10, 30, 100, 300, 1000.

Figure 11 show how smooth the functions of $\eta$, $f_{CP}$, CP area over $N_D$ and $V_{TH}^{EFF}$ are when $V_{OP}$ is 0.25 V. $V_{TH}^{EFF}$ is 0.03 V in Figure 11a–c. $N_D$ is 30 in Figure 11d–f. The arrows in Figure 11a,c indicate the optimum design plotted in Figure 10e. As $N_D$ increases, CP can run faster to keep $V_{TH}^{EFF}$, as shown in Figure 11b. To obtain a target output current at a target output voltage, capacitors can be scaled with $f_{CP}$ in SSL, resulting in scaled CP area with larger $N_D$, as shown in Figure 11c. Faster operation increases the current for top and bottom parasitic capacitances, resulting in less power efficiency, as shown in Figure 11a. Similar tendencies are valid for the sensitivities of $\eta$, $f_{CP}$, CP area on $V_{TH}^{EFF}$. To reduce the voltage difference between the next neighbor stages at the falling edge, $f_{CP}$ needs to be lower, as shown in Figure 11e. As a result, $\eta$ and CP area decreases as $V_{TH}^{EFF}$ increases, as shown in Figure 11d,f, respectively.

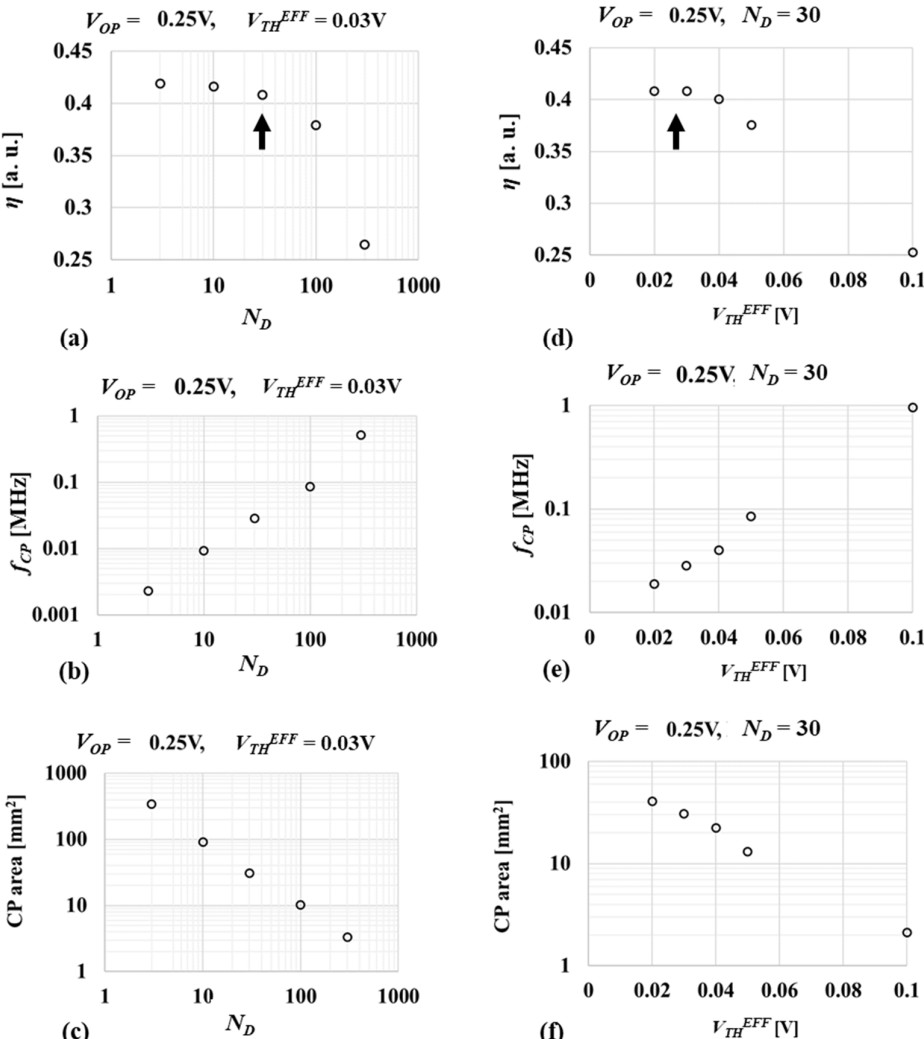

**Figure 11.** (**a**) $\eta$, (**b**) $f_{CP}$, (**c**) CP area vs. $N_D$ and (**d**) $\eta$, (**e**) $f_{CP}$, (**f**) CP area vs. $V_{TH}^{EFF.}$

Figure 12a shows the relative design parameter values normalized by the values at Vop = 0.75 V, which are $N_D$ = 300, $N_{CP}$ = 5, $C_{CP}$ = 1.1 nF, $f_{CP}$ = 113 kHz, $\alpha_T$ = 2.9 %, $V_{TH}^{EFF}$ = 40 mV, $A_{CP}$ = 0.62 mm², $I_{OP}$ = 240 µA, $\eta$ = 0.50. Capacitance per stage and CP area have strong Vop dependence except for the glitches at Vop = 1.0 V, as explained above on Figure 10b. Higher Vop is generally required to have small CP for cost reduction. Figure 12b shows the input current of CP, $I_{OP}$, when the CP is designed to run at the input voltage of $V_{OP}$ to output $I_{PP}$ at $V_{PP}$ with the high $\eta$. The slope was about −1.16 like the curve (c) of Figure 4, which indicates that a higher $V_{OP}$ basically allows a smaller TEG.

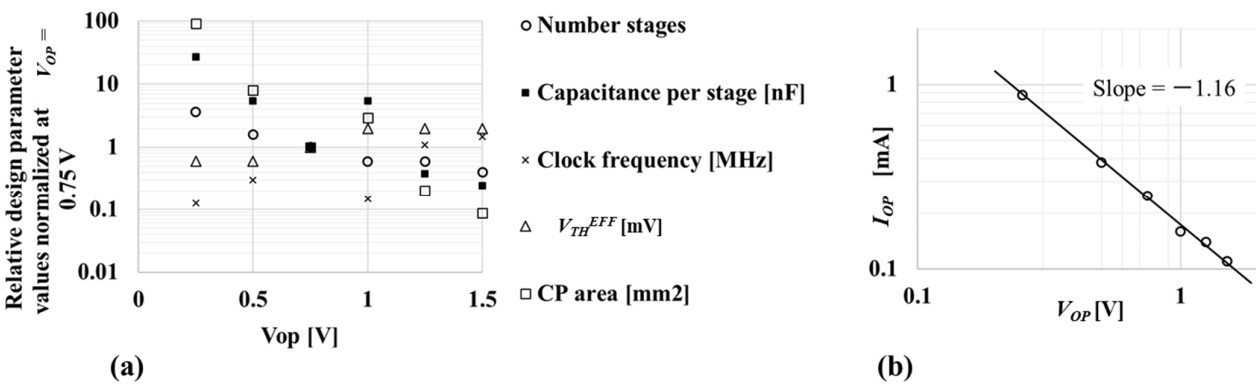

**Figure 12.** Trend of optimum design parameters (**a**) and $I_{OP}$ (**b**) across *Vop*.

Figure 13a shows $\eta$ of CP vs. V$_{OP}$. CP1's were the optimized designs as shown by the bold arrows in Figure 10a–e. CP2 indicates another design with 6% lower $\eta$ and 90% smaller area at $V_{OP}$ = 1 V shown by the broken arrow in Figure 10b. $\eta$ tends to increase as V$_{OP}$. Figure 13b shows TEG area as a function of $V_{OP}$ by using (9) or (11) for the CPs depending on whether a variable $V_{OC}$ range is unlimited or limited. Equation (9) is valid across the entire $V_{OP}$ range in case of $V_{OC\_MAX} \geq 3$ V whereas (11) is used when $V_{OP} \geq 0.8$ V in case of $V_{OC\_MAX}$ = 1.6 V. TEG can be minimized at a higher $V_{OP}$ when $V_{OC\_MAX} \geq 3$ V because CP nominally has a higher $\eta$ at a higher $V_{OP}$. On the other hand, when $V_{OC\_MAX}$ is limited, $V_{OP}$ around $V_{OC\_MAX}/2$ provides the minimum area for TEG. In this demonstration, $V_{OP}$ to have TEG area as small as minimum is 1.0 V with CP1 or between 0.5 V and 0.75 V with CP1 or 1.0 V with CP2. Figure 13c shows CP area as a function of $V_{OP}$. Basically, CP area exponentially increases as $V_{OP}$ decreases. When $V_{OC\_MAX}$ is limited at 1.6 V, the minimum TEG cost is realized with CP1 operated at 1.0 V. CP1 area is about 1.0 mm². If 10% larger TEG cost is acceptable, CP2 with 0.1 mm² would be another option. Thus, once the actual operating point $V_{OP}$ and $I_{OP}$ are determined based on such graphs as Figure 13b, c, one can design TEG based on (22) or (26) under the condition that $R_{TC}$ and the minimum $V_{TC}$ of a unit TEG are given, depending on the $V_{OC\_MAX}$ condition as discussed above.

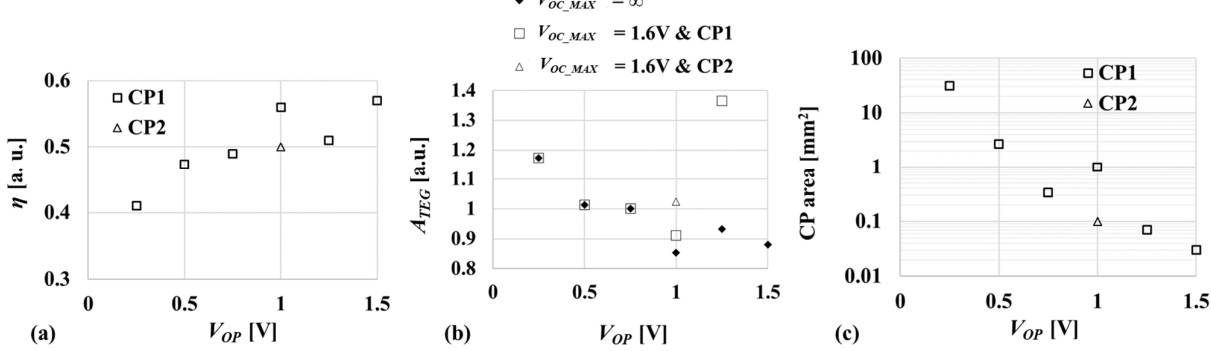

**Figure 13.** $\eta$ of CP (**a**), $A_{TEG}$ (**b**) and CP area (**c**) vs. $V_{OP}$.

In summary, CP design flow is as follows:

(1) The minimum required output current $I_{PP\_TGT}$ at the target output voltage $V_{PP}$ are specified by the load.
(2) The optimum CP is designed to have the minimum input power as a function of the input voltage $V_{OP}$ based on equations (9) through (21).
(3) The results provide the required TEG output current $I_{OP}$ at every $V_{OP}$.

TEG design flow is then as follows:

(4) The minimum temperature difference in operation is specified, which determines the output impedance RTC and open circuit voltage VTC of a TEG unit.

(5) The number of TEG arrays NP and the number of TEG units connected in series per array NS are determined to minimize the TEG area, i.e., the TEG cost, based on equations (22) through (26).

To see if the CP design flow using Table I is sufficiently valid, the gate-level CP2 circuit to operate at $V_{OP}$ of 1.0 V was designed in 65 nm CMOS. Ultra-low-power diodes [20] were used for switching diodes. The CP was simulated together with TEG whose $V_{OC}$ and $R_{TEG}$ were 1.6 V and 2.5 kΩ, respectively. The $V_{PP}$–$I_{PP}$ curve of the model was in good agreement with SPICE simulation as shown in Figure 14.

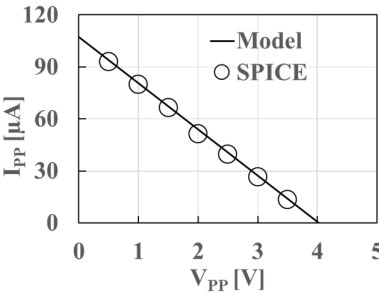

**Figure 14.** $V_{PP}$–$I_{PP}$ of CP2 operating with TEG whose $V_{OC}$ and $R_{TEG}$ are 1.6 V and 2.5 kΩ.

When the parasitic resistance of the interconnection to connect multiple TEG units is not negligibly small or the oscillator cell consumes substantial power, proper corrections would need to be done to accurately design the TEG–CP system with minimum cost.

## 4. Conclusions

A practical design flow for minimizing TEG energy harvester was proposed and demonstrated taking interaction between the TEG electrical parameters such as the open circuit voltage and output resistance of TEG and the load conditions such as the input voltage and current of sensor/RF chip and the power conversion efficiency of the Dickson charge pump converter in autonomous sensor modules into consideration. By using the proposed design flow, one can determine the total number of TEG units together with the number of TEG arrays and the number of TEG units connected in series per array for minimum TEG cost.

**Author Contributions:** Conceptualization, T.T.; methodology, K.K. and T.T.; software, K.K.; validation, K.K. and T.T.; formal analysis, K.K. and T.T.; investigation, K.K. and T.T.; writing—original draft preparation, K.K.; writing—review and editing, T.T.; funding acquisition, T.T. All authors have read and agreed to the published version of the manuscript.

**Funding:** This research was funded by Zeon Corp.

**Data Availability Statement:** Not applicable.

**Conflicts of Interest:** The authors declare no conflict of interest.

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
