# Peer review of "A Design of a Thermoelectric Energy Harvester for Minimizing Sensor Module Cost"

_electronics, doi:10.3390/electronics11213441_

Round 1

Reviewer 1 Report

The problem of the relationship between thermoelectric generator (TEG) electrical parameters, power efficiency of converters, and power consumption of loads is interesting and remain unadressed in the reseach filed of TEG systems. Hence, the study topic of the present work is very interesting. However, I cannot find any new findings and any solutions to abovementioned problem. All the equations and conclusions are konwn already after a few year's working in the TEG field. Several observations are as follows

(1) After Equation (4), the Rteg also related to running temperature. Therefore, this mistake must be corrected.

(2) Actually, no one can predicet the power consumption of loads. Usually, a battery or a capacitance is installed between TEG system and loads.

(3) I cannot find the method to "determine the total number of TEG units together with the number of TEG arrays and the number of TEG units connected in series per array when the characteristics of TEG unit, the minimum temperature difference in operation, the power conversion efficiency of the converter and the load condition are give". Moreover, the results presented can not suport such conculsion.

Reviewer 2 Report

 The paper “A Design of Thermoelectric Energy Harvester for Minimizing Sensor Module Cost” is in the conference format. The article is written in a tutorial note useful in design and from this point of view is consistent. Unfortunately, it is too close to a technical note to be considered for publication in this form. It is an article from which readers can learn a lot but the novelty elements are missing or are vague. Given the limited bibliography and a rearrangement of the figures horizontally with two figures on the line (a), (b) would significantly reduce the space occupied by the article somewhere altogether to about 7-8 pages. Unfortunately, this article is not the best in the famous career of the authors but rather below average. Elements specific to an article must be added (modeling, simulation, experiment, novelty elements, innovation in the investigated case). A lot of growth at work and I'm sure that in the next version many elements will be significantly improved.

Reviewer 3 Report

This paper discusses a relationship between thermoelectric generator (TEG) electrical parameters, power efficiency of converters, and power consumption of loads in autonomous sensor modules. Based on the method discussed, one can determine the total number of TEG units together with the number of TEG arrays and the number of TEG units connected in series per array when the characteristics of TEG unit, the minimum temperature difference in operation, the power conversion efficiency of the converter and the load condition are given. A practical design flow to minimize TEG cost is proposed and demonstrated, taking the maximum open circuit voltage of TEG and the dependence of the power conversion efficiency of the converter on the input voltage of the converter into consideration. 

The work is interesting and useful for energy harvesting applications present in the modern interconnected era. The paper is well-written and organized. English proof-reading is strongly suggested, honestly speaking in some part of the paper the sentences are not so clear. The paper is suitable for publication on this journal. My comments are reported in the following.

1 - Paper should include more relevant references, looking at the CP (as the reported example of power converter mainly used in monolithic implementations) a good number of works are introduced in literature, which could be cited to enhance the introduction. To give some example,

          - I. Doms, P. Merken, C. Van Hoof, and R. P. Mertens, “Capacitive Power Management Circuit for Micropower Thermoelectric Generators With a 1.4 μA Controller,” IEEE J. Solid-State Circuits, vol. 44, no. 10, pp. 2824–2833, Oct. 2009, doi: 10.1109/JSSC.2009.2027546.

- P. Weng, H. Tang, P. Ku and L. Lu, "50 mV-Input Batteryless Boost Converter for Thermal Energy Harvesting," IEEE J. Solid-State Circuits, vol. 48, no. 4, pp. 1031-1041, Apr. 2013, doi: 10.1109/JSSC.2013.2237998.

- A. Ballo, A. D. Grasso and G. Palumbo, "A Subthreshold Cross-Coupled Hybrid Charge Pump for 50-mV Cold-Start," in IEEE Access, vol. 8, pp. 188959-188969, 2020, doi: 10.1109/ACCESS.2020.3032452.

- S. Bose, T. Anand, and M. L. Johnston, “Integrated Cold Start of a Boost Converter at 57 mV Using Cross-Coupled Complementary Charge Pumps and Ultra-Low-Voltage Ring Oscillator,” IEEE J. Solid-State Circuits, -vol. 54, no. 10, pp. 2867–2878, Oct. 2019, doi: 10.1109/JSSC.2019.2930911.

-  M. Dezyani, H. Ghafoorifard, S. Sheikhaei, and W. A. Serdijn, “A 60 mV Input Voltage, Process Tolerant Start-Up System for Thermoelectric Energy Harvesting,” IEEE Trans. Circuits Syst. Regul. Pap., vol. 65, no. 10, pp. 3568–3577, Oct. 2018, doi: 10.1109/TCSI.2018.2834312.

2 – It should be specified if the CP is assumed to work in slow switching limit (SSL), as I think it operates, or in fast switching limit (FSL). In each case, does anything change if the CP operates in the opposite limit?

3 – Are the reported results measured or simulated? If they are measured, a picture of the measurement setup is needed.

4 – An example of transistor level implementation should be shown to better understand what CP is used. As an example, it is strongly different to apply a Fibonacci CP rather than a Dickson one (output-to-input voltage ratio as well as the output impedance differ for both topologies).

5 – There are aspects, not taken into consideration in this paper, that could affect the achieved results? For example, second order non-idealities of the single TEG and interconnection among the various sub cells, and for the CP as well? For the last, I suppose that first and second order non-idealities are gathered in the single parameter eta (power efficiency), isn’t it ?

6 - Finally, there are other type of optimization method for multi-TEG systems in literature ? If yes, a comparison with the other methods should be done.

Round 2

Reviewer 1 Report

The authors do not address my concerns properly.

Author Response

The following sentences were added in the latest revised manuscript. 

In summary, CP design flow is as follows:

1)The minimum required output current Ipp_tgt at the target output voltage Vpp are specified by the load.

2)The optimum CP is designed to have the minimum input power as a function of the input voltage Vop based on equations (9) through (21).

3)The results provide the required TEG output current Iop at every Vop.

And then, TEG design flow is as follwos:

4)    The minimum temperature difference in operation is specified, which determines the output impedance RTC and open circuit voltage VTC of a TEG unit.

5)    The number of TEG arrays Np and the number of TEG units connected in series per array Ns are determined to minimize the TEG area, i.e., the TEG cost, based on equations (22) through (26).

Reviewer 2 Report

In the new version of the paper the authors improve the article in direction suggested in previous review. 

Author Response

Thank you for reviewing the revised manuscript. 

Reviewer 3 Report

In my opinion, the manuscript can be accepted for pubblication in this journal.

Author Response

(The authors gave the same response as above.)
